# Discovery of Extended Summary Graphs in Time Series

**Charles K. Assaad**[1,2]  **Emilie Devijver**[2]  **Eric Gaussier**[2]

[1]EasyVista, 38000, Grenoble, France
[2]Univ. Grenoble Alpes, CNRS, Grenoble INP, LIG, 38000 Grenoble, France

## Abstract

This study addresses the problem of learning an extended summary causal graph from time series. The algorithms we propose fit within the well-known constraint-based framework for causal discovery and make use of information-theoretic measures to determine (in)dependencies between time series. We first introduce generalizations of the causation entropy measure to any lagged or instantaneous relations, prior to using this measure to construct extended summary causal graphs by adapting two well-known algorithms, namely PC and FCI. The behaviour of our method is illustrated through several experiments.

## 1 INTRODUCTION

Time series arise as soon as observations, from sensors, for example, are collected over time. They are present in various forms in many different domains, as healthcare (through, *e.g.*, monitoring systems), Industry 4.0 (through, *e.g.*, predictive maintenance and industrial monitoring systems), surveillance systems (from images, acoustic signals, seismic waves, etc.) or energy management (through, *e.g.* energy consumption data) to name but a few. We are interested in this study in analyzing time series to detect the causal relations that exist between them. In other words, we aim to build a causal graph from observational data[1]. In such graphs, nodes represent variables, in our case the time series or their evaluation onto timepoints, and arrowheads represent the direction of the causal relation, from causes to effects. Different types of causal graphs can be considered for time series: full-time causal graphs which cover all time instants, window causal graphs (Figure 1 (a)) which only cover a fixed number of time instants, summary causal graphs (Figure 1 (b)) which

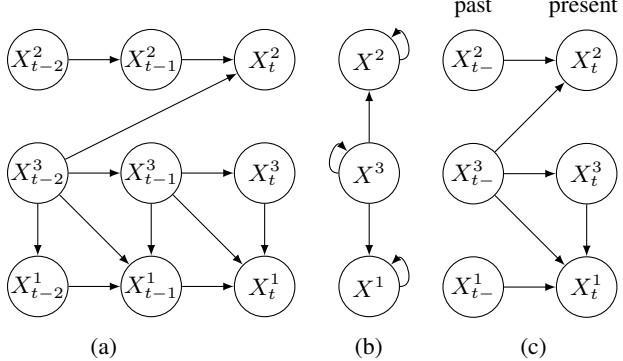

Figure 1: Example of a window causal graph (a), a summary causal graph (b) and an extended summary causal graph (c).

directly relate variables without any indication of time [Assaad et al., 2022].

Considering a full-time causal graph is not realistic for long time series; furthermore, when causal relations are consistent through time, a property known as causal stationarity, full-time causal graphs reduce to window causal graphs, the size of the window being given by the largest time gap $\gamma$ between causes and effects. This said, it is difficult for an expert to provide a window causal graph because it is difficult to determine which exact time instant is the cause of another. It is of course easier for an expert to propose a summary causal graph. However, such a summary hides the temporal relations between variables in the sense that a causal relation (excluding self causes) can either be instantaneous or relate time series at different time instants. To address this problem, we consider in this study *extended summary causal graphs* (Figure 1 (c)) in which past instants are conflated in a *past slice* and present instants represented in a *present slice*. Extended summary causal graphs can represent two types of relations[2]: from the past (represented

---

[1]We use here the term *observational data* to refer to observed data on which one cannot intervene.

[2]As such, they are similar to the two time-slice Bayesian networks [Koller and Friedman, 2009].

*Accepted for the 38th Conference on Uncertainty in Artificial Intelligence* (UAI 2022).

for a time series $X^p$ by $X^p_{t-}$) to the present (represented for a time series $X^p$ by $X^p_t$) and instantaneous relations in the present slice.

Potential effects in extended summary graphs are variables in the present slice, whereas potential causes are variables in both the past and present slices, as illustrated in Figure 1 (c). Lastly, as the underlying full-time graph is acyclic, both window causal graphs and extended summary causal graphs are also acyclic. This is not necessarily the case for summary causal graphs.

Previous studies have investigated methods to build window causal graphs from observational data, from which extended summary causal graphs and summary causal graphs can be directly deduced [Entner and Hoyer, 2010, Hyvärinen et al., 2010, Nauta et al., 2019, Runge et al., 2019, Runge, 2020]. However, this process is costly as one needs to explicitly identify all causal relations between any two pairs of time series. Furthermore, methods directly aiming at building (extended) summary graphs may be more robust to noise and finally more precise for these particular graphs. In addition, there are several situations in which one is mainly interested in the (extended) summary graphs as these graphs provide a simple, yet operational, view on the causal relations that exist between time series. Lastly, as argued before, contrary to (extended) summary causal graphs, window causal graphs may be difficult to analyze by experts.

Our goal here is to provide efficient procedures to directly build extended summary causal graphs. We do so by exploiting the causal relations in the window causal graphs without explicitly stating them. More explicitly, our contributions are two-fold:

1. first, we introduce a new information measure referred to as *greedy causation entropy* that can help in detecting if a past slice of a time series is independent or conditionilly independent of the present slice of another time series;

2. then, we combine this measure with a PC-based algorithm [Spirtes et al., 2000] for causal discovery with no hidden common causes, and with a FCI variant [Spirtes et al., 2000, Zhang, 2008] for causal discovery with hidden common causes. In both cases, the orientation rules are adapted to extended summary causal graphs.

The remainder of the paper is organized as follows: Section 2 presents the related work; the greedy causation entropy is introduced in Section 3 and the causal discovery algorithms in Section 4. Section 5 describes the experiments conducted to evaluate our proposal and Section 6 concludes the paper.

## 2 RELATED WORK

Granger Causality is one of the oldest methods proposed to detect causal relations between time series. However, in its standard form [Granger, 1969], it is known to handle a restricted version of causality that focuses on linear relations and causal priorities as it assumes that the past of a cause is necessary and sufficient for optimally forecasting its effect. This approach has nevertheless been improved since then [Granger, 2004, Arnold et al., 2007] through, *e.g*, the use of variable selection tools. Recently, Granger causality has been explored through an attention mechanism within convolutional networks [Nauta et al., 2019] to handle non linear relations and, in special cases, hidden common causes.

In a different line, approaches based on Structural Equation Models assume that the causal system can be defined by a set of equations that explain each variable by its direct causes and an additional noise. Causal relations are in this case discovered using footprints produced by the causal asymmetry in the data. For time series, the most popular algorithms in this family are VarLiNGAM [Hyvärinen et al., 2008], which is an extension of LiNGAM through autoregressive models, and TiMINo [Peters et al., 2013], which discovers a causal relationship by looking at independence between the noise and the potential causes. The main drawbacks of these approaches are the need of a larger sample size to achieve a performance comparable to other classes of methods, as well as the simplifying assumptions made on the relations between causes and effects [Malinsky and Danks, 2018].

Score-based approaches [Chickering, 2002] search over the space of possible graphs trying to maximize a score that reflects how well the graph fits the data. Recently, a new score-based method called Dynotears [Pamfil et al., 2020] was presented to infer a window causal graph from time series. However, it was shown that this method aim at finding sparse structural equation models that best explain the data, without any guarantee on the corresponding DAG [Kaiser and Sipos, 2021].

Constraint-based approaches, based on the PC algorithm by Spirtes et al. [2000], are certainly one of the most popular approaches for inferring causal graphs. Several algorithms, adapted from non-temporal causal graph discovery algorithms, have been proposed in this family for time series, among which oCSE by Sun et al. [2015] and PCMCI by Runge et al. [2019], Runge [2020] which aims to infer a window causal graph and uses standard mutual information to assess whether two variables are causally related or not. Other variants such as tsFCI by Entner and Hoyer [2010], SVAR-FCI by Malinsky and Spirtes [2018], and LPCMCI by Gerhardus and Runge [2020] focus on hidden common causes. Our work fits within this family, but we focus here on the extended summary causal graph and introduce a specific entropy measure for that purpose.

The application of information theoretic measures to temporal data raises several problems due to the fact that time series can be shifted in time and may have strong internal

dependencies. Many studies have attempted to re-formalize mutual information for time series: Galka et al. [2006] decorrelated observations by whitening data (which may have severe consequences on causal relations); Schreiber [2000] represents the information flow from one state to another with an asymmetric transfer entropy measure; Frenzel and Pompe [2007], inspired by Kraskov et al. [2004], represented time series by vectors that are assumed to be statistically independent; the Time Delayed Mutual Information proposed in Albers and Hripcsak [2012] aims at addressing the problem of non uniform sampling rates. The measure we propose bears some similarity with transfer entropy as it is also asymmetric; it is however suited to discover extended summary graphs as it can consider potentially complex relations between timestamps in different time series through the use of windows.

## 3 GREEDY CAUSATION ENTROPY

Assuming a relaxed version of *temporal priority*, which states that effects do not occur before their causes, and *consistency throughout time* or *causal stationarity*, which states that all causal relations remain constant throughout time, we consider the following general form for the functional causal model of any potential effect $X^q$:

$$\forall t,\, X_t^q = f(\mathcal{C}_t^q(X^{p_1}), \cdots, \mathcal{C}_t^q(X^{p_q}), \xi_t^q), \qquad (1)$$

where $f$ denotes any real-valued multivariate function and $\xi_t^q$ represent the noise terms which are serially and mutually independent of each other and are independent from all the causes of $X_t^q$. $\mathcal{C}_t^q(X^p)$, for $X^p$ a cause of $X_t^q$, represents the past and present instants (*i.e.*, time instants before $t$ or equal to $t$) of $X^p$ which are direct causes of $X_t^q$; it can be written as:

$$\mathcal{C}_t^q(X^p) = \{X_{t-\gamma_1}^p, \cdots, X_{t-\gamma_{K_r}}^p\},$$

where $K_r \in \mathbb{Z}^+$ and $\gamma_1, \cdots, \gamma_{K_r}$ are integers such that $\gamma_1 > \cdots > \gamma_{K_r} \geq 0$. As past instants of a time series can be causes of its present instant, $X^q$ can of course be a cause of itself.

The general functional model of Eq. 1 shows that the causal relation between a cause $X^p$ and its effect $X^q$ is captured through the relation between $X_t^q$ and its direct causes in $X^p$. If one measures (in)dependence with mutual information, denoted $I$ in the remainder, then one can conclude that $X^p$ does not directly cause $X^q$ if one has, $\forall K \in \mathbb{Z}^*$, $\forall \{\gamma_1, \cdots, \gamma_K\}$ s.t $0 \leq \gamma_K < \cdots < \gamma_1$:

$$I(X_t^q; X_{t-\gamma_1}^p, \cdots, X_{t-\gamma_K}^p) = 0.$$

The above statement can of course be extended by conditioning on any subset of past instants of any set of time series.

The computation of the above mutual information for all $K > 0$ and $\{\gamma_1, \cdots, \gamma_K\}$ can be time consuming as, for a given potential effect $X^q$ and potential cause $X^p$, its complexity is $\mathcal{O}(2^\gamma C_I)$, where $\gamma$ is the maximum gap between a cause and its effect and $C_I$ the complexity of the computation of the mutual information. It furthermore requires $\mathcal{O}(2^\gamma C_I)$ independence tests to assess whether the mutual information values obtained differ from 0 or not. Fortunately, the following property shows that one can still efficiently identify independence between $X^p$ and $X^q$ by considering the window in $X^p$ starting at $t - \gamma$ and ending at $t$, denoted $t-\gamma:t$.

**Property 1** *Let $\gamma$ denote the maximum gap between a cause and its effect. The following two propositions are equivalent:*

**(a)** $I(X_t^q; X_{t-\gamma_1}^p, \cdots, X_{t-\gamma_K}^p) = 0, \forall K \geq 1, \forall \gamma_1 > \cdots > \gamma_K \geq 0,$

**(b)** $I(X_t^q; X_{t-\gamma:t}^p) = 0.$

*The same equivalence holds for the conditional mutual information, using any conditional set.*

**Proof** Using the chain rule of mutual information, one has for all $K > 0$ and $\Gamma = \{\gamma_1, \cdots, \gamma_K\}$ such that $0 \leq \gamma_K < \cdots < \gamma_1$:

$$\begin{aligned} I(X_t^q; X_{t-\gamma:t}^p) =& I(X_t^q; X_{t-\gamma_1}^p, \cdots, X_{t-\gamma_K}^p) \\ &+ I(X_t^q; X_{(t-\gamma:t)\setminus\Gamma}^p | X_{t-\gamma_1}^p, \cdots, X_{t-\gamma_K}^p), \end{aligned}$$

where $X_{(t-\gamma:t)\setminus\Gamma}^p$ represents all time instants in $X_{t-\gamma:t}^p$ but $t-\gamma_1, \cdots, t-\gamma_K$. As mutual information is always positive, one can see that the left-hand side of the above equality is greater than or equal to the first term in the right-hand side of the equality, which shows that $(b) \Rightarrow (a)$. Furthermore, as $(a)$ is true for all $K$ and $\Gamma$, $(a) \Rightarrow (b)$. $\qquad\square$

Using the mutual information in (b) reduces the complexity of computing $I(X_t^q; X_{t-\gamma_1}^p, \cdots, X_{t-\gamma_K}^p)$ for all $K$ and all subsets of $K$ past instants in $X^p$ to $\mathcal{O}(C_I)$ and a single independence test.

The extended summary graph differentiates past and present instants of a time series, such that each time series is represented by two variables, as illustrated in Figure 1(c). The relations between time series in the present slice correspond to instantaneous relations. The standard (conditional) mutual information, $I(X_t^q; X_t^p)$, with complexity $\mathcal{O}(C_I)$, can be readily used to assess whether variables in the present slice are (conditionally) causally related or not, where the conditional set might be in the present or past slices. We will see below how to orient edges identified in the present slice.

To assess whether there exist causal relations between variables in the past and potential effect in the present slices,

we make use of the following *greedy causation entropy*[3] which is based on Prop. 1 and is asymmetric to reflect the specific role of the cause and the effect. Relations between variables in the past and present slices are naturally oriented by temporal priority.

**Definition 1** *With the same notations as before, the* greedy causation entropy, *denoted by GCE, from the time series $X^p$ to the time series $X^q$ is defined by:*

$$GCE(X^p \to X^q) = I(X_t^q; X_{t-\gamma:t-1}^p). \qquad (2)$$

*Denoting by $X^{\mathbf{Pr}}$ a set of $m$ time series $\{X^{Pr_1}, \cdots, X_t^{Pr_m}\}$ in the present slice and by $X^{\mathbf{Pa}}$ a set of $l$ time series $\{X_{t-}^{Pa_1}, \cdots, X_{t-}^{Pa_l}\}$ in the past slice, the* conditional greedy causation entropy *furthermore takes the form:*

$$GCE(X^p \to X^q | X^{\mathbf{Pa}}, X^{\mathbf{Pr}}) \qquad (3)$$
$$= I(X_t^q; X_{t-\gamma:t-1}^p | X_{t-}^{Pa_1}, \cdots, X_{t-}^{Pa_l}, X_t^{Pr_1}, \cdots, X_t^{Pr_m}).$$

Because of Prop. 1, one can conclude that past instants of $X^p$ do not directly cause $X^q$ iff there exists $X^{\mathbf{Pr}} = \{X_t^{Pr_1}, \cdots, X_t^{Pr_m}\}$ and $X^{\mathbf{Pa}} = \{X_{t-}^{Pa_1}, \cdots, X_{t-}^{Pa_l}\}$, with $m, l \geq 0$, such that $GCE(X^p \to X^q | X^{\mathbf{Pa}}, X^{\mathbf{Pr}}) = 0$. In the following, for simplification purposes, we will not differentiate in the conditioning time series in the present and past slices and will simply write $GCE(X^p \to X^q | X^R)$. Lastly, note that for determining (in)dependencies in the present slice, we directly rely on the standard (conditional) mutual information.

### 3.1 ESTIMATION

We rely on the $k$-nearest neighbor method [Frenzel and Pompe, 2007] for the estimation of standard mutual information. We present its adaptation to $GCE(X^p \to X^q | X^{\mathbf{R}})$ for $X^{\mathbf{R}}$ a set of $m$ time series $\{X^{r_1}, \cdots, X^{r_m}\}$. First, the distance we consider between two pairs of observations $i$ and $j$ is the supremum distance:

$$d((X_t^q, X_{t-\gamma:t-1}^p)_i, (X_t^q, X_{t-\gamma:t-1}^p)_j)$$
$$= \max\left(|(X_t^q)_i - (X_t^q)_j|, \max_{1 \leq \ell \leq \gamma} |(X_{t-\ell}^p)_i - (X_{t-\ell}^p)_j|\right).$$

Let us denote by $\epsilon_{ik}/2$ the distance from $(X_t^q, X_{t-\gamma:t-1}^p, X^{\mathbf{R}})_i$ to its $k$-th neighbor, and $n_i^{1,3}$, $n_i^{2,3}$ and $n_i^3$ the numbers of points with distance strictly smaller than $\epsilon_{ik}/2$ for the examples $(X_t^q, X^{\mathbf{R}})_i$, $(X_{t-\gamma:t-1}^p, X^{\mathbf{R}})_i$ and $(X^{\mathbf{R}})_i$. The estimate of the greedy causation entropy is

---

[3]We call it greedy because it considers all past instants (up to $\gamma$) without trying to filter them.

then given by:

$$\widehat{GCE}(X^p; X^q \mid X^{\mathbf{R}})$$
$$= \psi(k) + \frac{1}{n} \sum_{i=1}^n \psi(n_i^3) - \psi(n_i^{1,3}) - \psi(n_i^{2,3}),$$

where $\psi$ denotes the digamma function.

## 4 CAUSAL DISCOVERY FOR EXTENDED SUMMARY GRAPHS

We make use of the PC algorithm to construct extended summary graphs from observational time series. The first step in PC consists in constructing a skeleton that relates causes and effects. Once this is done, the skeleton is oriented. We extend this to data with hidden common causes using an extension of the FCI algorithm.

### 4.1 SKELETON CONSTRUCTION

One first constructs an extended summary graph in which there is an edge from all time series in the past slice to all time series in the present slice and all time series in the present slice are connected to one another (not oriented). Each edge between $X^p$ in the past slice to $X^q$ in the present slice is then removed if $GCE(X^p \to X^q) = 0$. The same is done for the edges in the present slice using the usual mutual information. One then checks, for the remaining edges, whether the two time series are conditionally independent (the edge is removed) or not (the edge is kept). Starting from a single time series connected to $X^p$ or $X^q$, the set of conditioning time series is gradually increased till either the edge between $X^p$ and $X^q$ is removed or all time series connected to $X^p$ and $X^q$ have been considered, in both directions. The conditional version of GCE is used for edges between the past and present slices, whereas the conditional mutual information is used for edges in the present slice. In this procedure, we use the same strategy as the one used in PC-stable [Colombo and Maathuis, 2014] which consists in sorting time series according to their GCE or mutual information scores and, when an independence is detected, in removing all other occurrences of the time series. This leads to an order-independent procedure.

### 4.2 ORIENTATION UNDER CAUSAL SUFFICIENCY

We first assume that the set of observed time series is *causally sufficient* [Spirtes et al., 2000], that is all common causes of all time series are observed.

As noted before, the orientation of the edges between the past and present slices is straightforward. It is based on the temporal priority principle which states that an effect

cannot precede a cause. All these edges are thus oriented from the past to the present. We then try to orient as many edges as possible in the present slice by using standard PC rules which are applied recursively till no more edges can be oriented. The origin of causality and propagation of causality make use of both time series in the past and present slices as colliders can involve time series in the present and in the past slices. We give below the form the PC rules take in our case, where $\texttt{Sepset}(p \leftrightarrow q)$ denotes the separation set of $X^p$ and $X^q$ according to the conditional mutual information and $\texttt{Sepset}(p \rightarrow q)$ the separation set of $X^p$ and $X^q$ according to GCE:

**PC-Rule 0 (Origin of causality)**

(i) *In an unshielded triple $X_t^p - X_t^r - X_t^q$, if $X_t^r \notin \texttt{Sepset}(p \leftrightarrow q)$, then $X_t^r$ is an unshielded collider: $X_t^p \rightarrow X_t^r \leftarrow X_t^q$.*

(ii) *In an unshielded triple $X_{t-}^q \rightarrow X_t^q - X_t^p$, if $X_t^q \notin \texttt{Sepset}(q \rightarrow p)$, then $X_t^q$ is an unshielded collider: $X_{t-}^q \rightarrow X_t^q \leftarrow X_t^p$.*

**PC-Rule 1 (Propagation of causality)** *In an unshielded triple $X_t^p \rightarrow X_t^r - X_t^q$ (resp. $X_{t-}^p \rightarrow X_t^r - X_t^q$), if $X_t^r \in \texttt{Sepset}(p \leftrightarrow q)$ then orient the unshielded triple as $X_t^p \rightarrow X_t^r \rightarrow X_t^q$ (resp. $X_{t-}^p \rightarrow X_t^r \rightarrow X_t^q$).*

**PC-Rule 2** *If there exist a direct path from $X_t^p$ to $X_t^q$ and an edge between $X_t^p$ and $X_t^q$, then orient $X_t^p \rightarrow X_t^q$.*

**PC-Rule 3** *Orient $X_t^p - X_t^q$ as $X_t^p \rightarrow X_t^q$ whenever there are two paths $X_t^p - X_t^r \rightarrow X_t^q$ and $X_t^p - X_t^s \rightarrow X_t^q$.*

As we are using here the standard PC rules, and under the faithfulness assumption [Spirtes et al., 2000], we have the following theorem, the proof of which directly derives from results on PC [Spirtes et al., 2000].

**Theorem 1 (Theorem 5.1 of Spirtes et al. [2000])** *Let the distribution of $V$ be faithful to a DAG $\mathcal{G} = (V, E)$, and assume that we are given perfect conditional independence information about all pairs of variables $(X^p, X^q)$ in $V$ given subsets $X^R \subseteq V \backslash \{X^p, X^q\}$. Then the skeleton constructed previously followed by the above orientation rules represents the CPDAG of of the extended summary causal graph $\mathcal{G}$.*

**Proof** Property 1 and GCE allow one to consider past instants of a given time series as a single meta-variable and to compute, through Eq. 3, conditional mutual information measures between such meta-variables and variables in the present slice. We are using d-separation and PC on these (meta-)variables; thus Theorem 5.1 applies when assuming

that the data distribution of the (meta-)variables is faithful to the extended summary graph. $\square$

The above theorem states that the construction procedure we have followed is correct and gives the completed partially directed acyclic graph (CPDAG) which corresponds to Markov equivalence class of the true causal graph [Andersson et al., 1997, Chickering, 2002]. The overall process is referred to as PCGCE and given in Algorithm 1.

---

**Algorithm 1** PCGCE

---

**Require:** $X$ a $d$-dimensional time series of length $T$, $\gamma \in \mathbb{N}$ the maximum number of lags, $\alpha$ a significance threshold
  **Initialization:** Construct a partially oriented extended summary graph $\mathcal{G} = (V = \{V_t, V_{t-}\}, E)$ with $2d$ nodes such that $\forall X_t^p, X_t^q \in V_t, X_t^p - X_t^q$ and $\forall X_{t-}^p \in V_{t-}, X_t^q \in V_t, X_{t-}^p \rightarrow X_t^q$
  n = 0
  **while** $\exists X_t^q \in V$ s.t. card$(\text{Adj}(X_t^q, \mathcal{G})) \geq n + 1$ **do**
    $\mathbf{D} = list()$
    **for** $X_t^q \in V_t$ s.t. card$(\text{Adj}(X_t^q, \mathcal{G})) \geq n + 1$ **do**
      **for** $X_{t*}^p \in \text{Adj}(X_t^q, \mathcal{G})$ such that $t^* \in \{t, t-\}$ **do**
        **for** all subsets $X^{\mathbf{R}} \subset \text{Adj}(X_t^q, \mathcal{G}) \backslash \{X_{t*}^p\}$ such that card$(X^{(\mathbf{R})}) = n$ **do**
          **if** $t^* = t$ **then**
            $y_{q,p,t,\mathbf{R}} = \text{I}(X^p; X^q \mid X^{\mathbf{R}})$
          **else**
            $y_{q,p,t-,\mathbf{R}} = \text{GCE}(X^p \rightarrow X^q \mid X^{\mathbf{R}})$
          append$(\mathbf{D}, \{X_t^q, X_{t*}^p, X^{\mathbf{R}}, y_{q,p,t^*,\mathbf{R}}\}))$
    Sort $\mathbf{D}$ by increasing order of $y$
    **while** $\mathbf{D}$ is not empty **do**
      $\{X_t^q, X_{t*}^p, X^{\mathbf{R}}, y\} = \text{pop}(\mathbf{D})$
      **if** $X_{t*}^p \in \text{Adj}(X_t^q, \mathcal{G})$ and $X^{\mathbf{R}} \subset \text{Adj}(X_t^q, \mathcal{G})$ **then**
        Compute $z$ the p-value of $y$ using a statistical independence test
        **if** $z > \alpha$ **then**
          **if** $t^* = t$ **then**
            Remove edge $X_t^p - X_t^q$ from $\mathcal{G}$
          **else**
            Remove edge $X_{t-}^p \rightarrow X_t^q$ from $\mathcal{G}$
          Sepset$(p_{t*}, q_t) = $ Sepset$(q_t, p_{t*}) = X^{\mathbf{R}}$
    n=n+1
  **for** each triple in $\mathcal{G}$ **do** apply PC-Rule 0
  **while** an edge can be oriented **do**
    **for** each triple in $\mathcal{G}$ **do** apply PC-Rules 1, 2, 3
  **Return** the extended summary causal graph $\mathcal{G}$

---

## 4.3 EXTENSION TO HIDDEN COMMON CAUSES

When there exist unobserved variables that cause two variables of interest (*i.e.*, hidden common causes), an extended summary graph is not suitable to represent causal relations, and one needs to resort to maximal ancestor graphs (MAGs) and extended summary MAGs. An extended summary MAG behaves as the usual MAG [Richardson and Spirtes, 2002]

for time series in the present slice. In addition, there is a double arrow between a time series in the past slice and a time series in the present slice of two time series if there exists at least one hidden common cause between instants of the two time series.

The PC algorithm is not appropriate to deal with hidden common causes. Instead, one should use the FCI algorithm introduced in Spirtes et al. [2000] which infers a PAG (partial ancestral graph), which can contain up to six types of edges: undirected ($-$), single arrow ($\rightarrow$ or $\leftarrow$), double arrow ($\leftrightarrow$) corresponding to a hidden common cause, undirected on one side and undetermined on the other ($-\circ$ or $\circ-$), directed on one side and undetermined on the other ($\circ\rightarrow$ or $\leftarrow\circ$), and undetermined on both sides ($\circ-\circ$). In what follows, a $*$ is used to represent any of these types. We extend here the version of the algorithm presented in Zhang [2008] to time series and extended summary causal graphs.

From the skeleton obtained in Section 4.1, unshielded colliders are detected using the following rule:

**FCI-Rule 0 (Origin of causality)**

**(i)** *In an unshielded triple $X_t^p *-\circ X_t^r \circ-* X_t^q$, if $X_t^r \notin$ Sepset$(p \leftrightarrow q)$, then $X_t^r$ is an unshielded collider: $X_t^p *\!\!\rightarrow X_t^r \leftarrow\!* X_t^q$.*

**(ii)** *In an unshielded triple $X_{t-}^q *\!\!\rightarrow X_t^q \circ-* X_t^p$, if $X_t^q \notin$ Sepset$(q \rightarrow p)$, then $X_t^q$ is an unshielded collider: $X_{t-}^q *\!\!\rightarrow X_t^q \leftarrow\!* X_t^p$.*

From this, we construct the Possible-Dsep sets, defined as follows:

**Definition 2** *Let $X_{t*}^r$ denote a time series in either the past or present slice. $X_{t*}^r$ is in the Possible-Dsep set of $X_{t-}^p$ and $X_t^q$ (resp. $X_t^p$ and $X_t^q$) if and only if $X_{t*}^r$ is different from $X_{t-}^p$ (resp. $X_t^p$) and $X_t^q$ and there is an undirected path $U$ between $X_{t-}^p$ (resp. $X_t^p$) and $X_{t*}^r$ such that for every subpath $< X_{t*}^w, X_{t*}^s, X_{t*}^v >$ of $U$, either $X_{t*}^s$ is a collider on the subpath, or $X_{t*}^w$ and $X_{t*}^v$ are adjacent in the PAG.*

As elements of Possible-Dsep sets in a PAG play a role similar to the ones of parents in a DAG, additional edges are removed by conditioning on the elements of the Possible-Dsep sets, using the same strategy as the one given in Section 4.1. All edges are then unoriented and the FCI-Rule 0 is again applied as some of the edges of the unshielded colliders originally detected may have been removed by the previous step. Then, as in FCI, we apply the rules 1, 2, 3 and 4 introduced in Spirtes et al. [2000], and the rules 8, 9 and 10 introduced in Zhang [2008]. We do not included Rules 5, 6 and 7 from Zhang [2008] as these rules deal with selection bias, a phenomenon that is not present in the datasets we consider. Including these rules in our framework is nevertheless straightforward. The overall process, is referred to as FCIGCE.

## 5    EXPERIMENTS

We propose first an extensive analysis on simulated data, generated from basic causal structures; then we perform an analysis on a widely used simulated benchmark, namely FMRI (Functional Magnetic Resonance Imaging) which is often considered as a "realistic" benchmark.

**Data:** The artificial datasets correspond to seven extended summary causal graphs, extracted from window causal graphs, among which five are causally sufficient ($\mathring{4}t_{t=0}$, $4t_{t>0}$, $\mathring{4}t_{t>0}$, $4t_{t\geq0}$, $\mathring{4}t_{t\geq0}$) presented in Table 2a and two are non causally sufficient ($7t2h_{t>0}$, $7t\mathring{2}h_{t>0}$) presented in Table 2b. Causally sufficient structures comprise four observed times series whereas non causally sufficient structures contain seven observed time series and two hidden time series. The generating process of all datasets is the following: for all $q$, for all $t > 0$,

$$X_t^q = a_{t-1}^{qq} X_{t-1}^q + \sum_p a_{t-l}^{pq} f(X_{t-l}^p) + 0.1\xi_t^q,$$

where $0 \leq l \leq 2^4$, $a_t^{jq} \sim \mathcal{U}([-1;-0.1] \cup [0.1;1])$ for all $1 \leq j \leq d$, $\xi_t^q \sim \mathcal{N}(0,1)$ and $f$ is a non linear function chosen at random in {absolute value, tanh, sine, cosine}. From this, we generate datasets with different characteristics to illustrate the behaviour of different causal discovery methods. For all datasets, we consider time series with 1000 timestamps.

In the remainder, the notation $4t$ or $7t$ represents the number of time series in the dataset, $\circ$ above means that the time series is self causal, $2h$ means that there are two hidden common causes in the dataset, and the subscripts $t = 0$, $t > 0$ and $t \geq 0$ mean that all causal relations are instantaneous, with a strictly positive lag and with a positive lag. In $\mathring{4}t_{t=0}$, all causal relations between different time series are instantaneous and all time series are caused by their own past ($a_{t-1}^{qq} > 0$ and $a_{t-l}^{pq} = 0$). In $4t_{t>0}$ and $7t2h_{t>0}$, all causal relations have a lag $l > 0$ and none of the time series is caused by its own past ($a_{t-1}^{qq} = 0$ and $a_{t-l}^{pq} > 0$). In $\mathring{4}t_{t>0}$ and $\mathring{7}t2h_{t>0}$, all causal relations have a lag $l > 0$ and all time series are caused by their own past ($a_{t-1}^{qq} > 0$ and $a_{t-l}^{pq} > 0$). In $4t_{t\geq0}$, causal relations are either instantaneous or have a lag $l > 0$ and none of the time series is caused by its own past. Finally, in $\mathring{4}t_{t\geq0}$, causal relations are either instantaneous or have a lag $l > 0$ and all time series are caused by their own past. For each structure and for each setting, we generate 10 different datasets over which the performance of each method is averaged.

The FMRI (Functional Magnetic Resonance Imaging) benchmark contains BOLD (Blood-oxygen-level dependent) datasets for 28 different underlying brain networks[5] [Smith

---

[4]For datasets with positive lags, $l$ is randomly chosen in $\{1;2\}$; thus, for roughly half of the edges, the lag is 2.

[5]Original data: https://www.fmrib.ox.ac.uk/

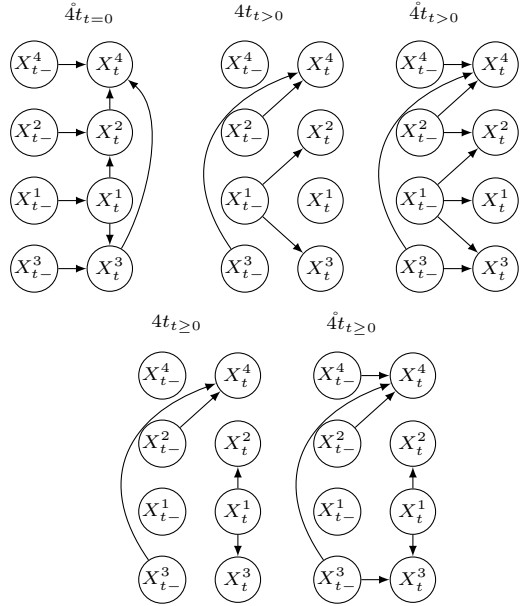

(a) Structures corresponding to the artificial datasets without hidden common causes. $A \rightarrow B$ means that A causes B.

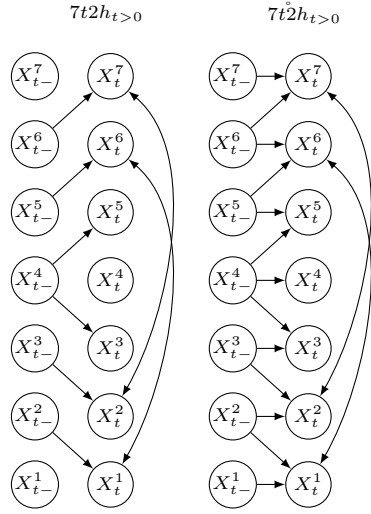

(b) Structures corresponding to the artificial datasets with hidden common causes. $A \rightarrow B$ means that A causes B and $A \longleftrightarrow B$ represents the existence of a hidden common cause between A and B.

Figure 2: Structures corresponding to the artificial datasets. The notation $4t$ or $7t$ represents the number of time series in the dataset, $\circ$ above means that the time series is self causal, $2h$ means that there are two hidden common causes in the dataset, and the subscripts $t = 0$, $t > 0$ and $t \geq 0$ mean that all causal relations are instantaneous, with a strictly positive lag and with a positive lag.

datasets/netsim/index.html
Preprocessed version: https://github.com/M-Nauta/TCDF/tree/master/data/fMRI

et al., 2011]. BOLD FMRI measures the neural activity of different regions of interest in the brain based on the change of blood flow. There are 50 regions in total, each with its own associated time series. Since not all existing methods can handle 50 time series (such as PCMCI using conditional mutual information and the associated permutation test), datasets with more than 10 time series are excluded. Furthermore, as the reference causal relations in the FMRI benchmark can only be represented by a summary causal graph, we compare all methods based on the summary causal graph they infer (this graph is directly deduced from the window causal graph or the extended summary causal graph for methods inferring these types of graphs).

**Methods:** All the methods retained can either infer a window causal graph, from which one can deduce the corresponding extended summary causal graph, or a summary causal graph with no instantaneous relations so that the extended summary causal graph can also be deduced (this is the case for oCSE and MVGCL presented below).

Among constraint-based methods, in addition to the proposed PCGCE and FCIGCE, we retained the well-known PCMCI[6] [Runge et al., 2019, Runge, 2020] which infers a window causal graph as well as oCSE [Sun et al., 2015], relying on our implementation, which infers an extended summary causal graph without instantaneous relations. For all those methods, the mutual information is estimated using the k-nearest neighbour method with $k$ fixed to 10; a significance local permutation test [Runge, 2018] with $k_{perm} = 5$ is furthermore used to assess whether the mutual information values differ from 0 or not. For non causally sufficient structures, we retained, in addition to FCIGCE, the state-of-the-art tsFCI[7] method [Entner and Hoyer, 2010] on which we use tests of zero correlation or zero partial correlation. The significance level of the test used is set to 0.05 for methods on causally sufficient structures (PCGCE, PCMCI, oCSE) and to 0.1 for methods on non causally sufficient structures (FCIGCE, tsFCI).

Among noise-based approaches, we retained the well-known VarLiNGAM[8] method [Hyvärinen et al., 2010], in which the regularization parameter in the adaptive Lasso is selected using the Bayesian Information Criterion (no statistical test is performed as we directly use the value of the statistics). From the Granger family, we retained the standard lasso-based multivariate Granger (GCMVL) [Arnold et al., 2007], which we re-implemented, and the recently proposed TCDF[9] [Nauta et al., 2019] with a kernel of size 4, a dilation coefficient set to 4, one hidden layer, a learning rate of 0.01, and 5000 epochs. Lastly, we retained, from

[6] https://github.com/jakobrunge/tigramite
[7] https://sites.google.com/site/dorisentner/publications/tsfci
[8] https://github.com/cdt15/lingam
[9] https://github.com/M-Nauta/TCDF

score-based approaches, the recently proposed Dynotears[10] method [Pamfil et al., 2020], the hyperparameters of which are set to their recommended values ($\lambda_W = \lambda_A = 0.05$ and $\alpha_W = \alpha_A = 0.01$).

For all the methods, we set the hyperparameter $\gamma$ to 5. A Python routine to use all the above methods is available at `https://github.com/ckassaad/PCGCE`.

**Evaluation Measures:** To assess the quality of causal inference, we use two different measures:

- $F^{p \neq q}$: the F1-score regarding causal relations between two different time series;
- $F^{p=q}$ : the F1-score regarding causal relations between a time series and itself.

**Results:** Table 1 summarizes the results of the different methods on causally sufficient simulated data. Overall, regarding causal relations between different time series (which are not linear due to the generation process retained), for all tested structures, PCGCE and PCMCI come out on top. In particular, PCGCE has the highest $F^{p \neq q}$ in the structures $\mathring{4}t_{t=0}$ and $4t_{t>0}$, followed by PCMCI and PCMCI has the highest $F^{p \neq q}$ in the structures $4t_{t \geq 0}$ and $\mathring{4}t_{t \geq 0}$, followed by PCGCE. In the structure $\mathring{4}t_{t>0}$ both methods PCGCE and PCMCI obtain the same $F^{p \neq q}$. oCSE is not evaluated on the structures $4t_{t=0}$, $4t_{t \geq 0}$ and $\mathring{4}t_{t \geq 0}$ since it cannot deal with instantaneous relations. However, for other structures, oCSE yields a low $F^{p \neq q}$ compared to other constraint-based methods (PCGCE and PCMCI), especially for the structure $4t_{t>0}$. For non constraint-based methods, MVGCL (which, as oCSE, cannot be evaluated on $4t_{t=0}$, $4t_{t \geq 0}$ and $\mathring{4}t_{t \geq 0}$) comes out best. On the other hand, Dynotears, VarLiNGAM and TCDF have poor performance. The results obtained with Dynotears, VarLiNGAM and MVGCL are expected as these methods are designed for linear relations (*i.e.*, in our case, self causes); in addition, VarLiNGAM is not capable of handling Gaussian noise. Regarding $F^{p=q}$, VarLiNGAM performs best for all structures followed by PCMCI and then by PCGCE. The difference in the results of VarLiNGAM in $F^{p \neq q}$ and $F^{p=q}$ is simply due to the fact that we considered non linear relations between two different time series but linear relations when the causal relations are within the same time series.

Table 2 summarizes the results obtained on the FMRI dataset using $F^{p \neq q}$ as the reference summary causal graph on this dataset does not contain self causes. As for simulated data, among constraint-based methods, PCGCE performs best with a $F^{p \neq q}$ significantly higher than the performance of PCMCI and oCSE. However, overall, for this dataset, non constraint-based methods, except TCDF, obtain better results. This suggests that the faithfulness assumption on which constraint-based methods rely, is not satisfied on

this dataset.

Lastly, we compare FCIGCE, tsFCI and TCDF on the two non causally sufficient structures described above in Table 3. For the first structure FCIGCE and tsFCI have the highest performance, FCIGCE being above tsFCI. For the second structure, tsFCI has the highest performance on both $F^{p \neq q}$ and $F^{p=q}$, followed by FCIGCE. TCDF performs poorly on both structures. We conjecture here that FCIGCE suffers from the use of a complete window when computing GCE, which can lead to less stable experimental results when the dataset is complex.

**Time complexity:** PC-based causal discovery algorithms (with instantaneous causal relations) have the following complexity, in terms of the number of independence tests [Spirtes et al., 2000], on window causal graphs: $(d(\gamma + 1))^2(d(\gamma + 1) - 1)^{k-1}/(2(k-1)!)$, where $d$ represents the number of time series considered. Algorithms adapted to time series, as PCMCI Runge [2020], rely on the assumption of temporal priority and consistency throughout time to reduce the number of tests. Our proposed method benefits from a smaller number of tests compared to PC and PCMCI if $\gamma > 1$. In the worst case, its complexity is: $4d^2(2d - 1)^{k-1}/(k-1)!$. However, our method needs to perform additional independence tests compared to oCSE as oCSE does not consider instantaneous causal relations. Figure 3 provides the computation computation of each constraint-based method on the causally sufficient structures. As one can note, PCGCE is slightly less efficient than oCSE and more efficient than PCMCI.

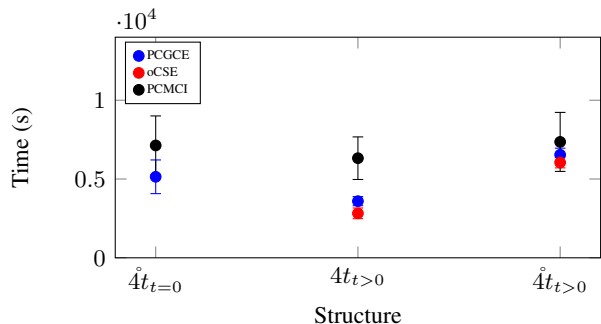

Figure 3: Time computation of constraint based algorithms on causally sufficient structures. oCSE is not computed on $4t_{t=0}$ as it does not consider instantaneous relations.

**Limitations and perspectives:** Since a cause in the past slice may contain up to $\gamma - 1$ dimensions, PCGCE can suffer when $\gamma$ increases, especially when the number of observations is fixed. To illustrate this, we reran the experiment for the structure $4\mathring{t}_{t \geq 0}$ with $\gamma$ set to 20. As expected, the F-scores ($F^{p \neq q}$ and $F^{p=q}$) of PCGCE decrease to $0.11 \pm 0.17$ and $0.8 \pm 0.12$, while other methods were able to maintain more or less the same F-scores (PCMCI: $0.57 \pm 0.18$ and $0.94 \pm 0.06$, VarLiNGAM: $0.19 \pm 0.12$ and $0.97 \pm 0.06$, Dynotears: $0.07 \pm 0.14$ and $0.37 \pm 0.21$, TCDF: $0.0 \pm 0.0$

[10]`https://github.com/quantumblacklabs/causalnex`

Table 1: Results for simulated data without hidden common causes. The mean and the standard deviation of the F1 score are reported and the best results are in bold. Double-bars are used for grouping methods according to the class they belong to.

| | Perf. | Constraint-based | | | Noise-based | Score-based | Granger-based | |
| | | PCGCE | oCSE | PCMCI | VarLiNGAM | Dynotears | TCDF | MVGCL |
|---|---|---|---|---|---|---|---|---|
| $\mathring{4}t_{t=0}$ | $F^{p \neq q}$ | **0.62** $\pm$ 0.17 | – | 0.60 $\pm$ 0.12 | 0.32 $\pm$ 0.13 | 0.04 $\pm$ 0.12 | 0.00 $\pm$ 0.00 | – |
| | $F^{p=q}$ | 0.81 $\pm$ 0.12 | – | 0.87 $\pm$ 0.12 | **0.92** $\pm$ 0.07 | 0.37 $\pm$ 0.21 | 0.18 $\pm$ 0.24 | – |
| $4t_{t>0}$ | $F^{p \neq q}$ | **0.71** $\pm$ 0.13 | 0.31 $\pm$ 0.21 | 0.67 $\pm$ 0.16 | 0.00 $\pm$ 0.00 | 0.16 $\pm$ 0.19 | 0.00 $\pm$ 0.00 | 0.52 $\pm$ 0.11 |
| $\mathring{4}t_{t>0}$ | $F^{p \neq q}$ | **0.81** $\pm$ 0.18 | 0.78 $\pm$ 0.17 | **0.81** $\pm$ 0.12 | 0.00 $\pm$ 0.00 | 0.16 $\pm$ 0.19 | 0.04 $\pm$ 0.12 | 0.53 $\pm$ 0.09 |
| | $F^{p=q}$ | 0.94 $\pm$ 0.06 | 0.82 $\pm$ 0.11 | 0.97 $\pm$ 0.05 | **0.98** $\pm$ 0.04 | 0.47 $\pm$ 0.15 | 0.35 $\pm$ 0.27 | – |
| $4t_{t \geq 0}$ | $F^{p \neq q}$ | 0.63 $\pm$ 0.13 | – | **0.69** $\pm$ 0.08 | 0.24 $\pm$ 0.21 | 0.14 $\pm$ 0.18 | 0.04 $\pm$ 0.12 | – |
| $\mathring{4}t_{t \geq 0}$ | $F^{p \neq q}$ | 0.54 $\pm$ 0.26 | – | **0.57** $\pm$ 0.20 | 0.19 $\pm$ 0.12 | 0.07 $\pm$ 0.15 | 0.04 $\pm$ 0.12 | – |
| | $F^{p=q}$ | 0.82 $\pm$ 0.11 | – | 0.94 $\pm$ 0.07 | **0.98** $\pm$ 0.04 | 0.37 $\pm$ 0.21 | 0.24 $\pm$ 0.30 | – |

Table 2: Results for realistic data. The mean and the standard deviation of the F1 score are reported and the best results are in bold. Double-bars are used for grouping methods according to the class they belong to.

| | Perf. | PCGCE | oCSE | PCMCI | VarLiNGAM | Dynotears | TCDF | MVGCL |
|---|---|---|---|---|---|---|---|---|
| FMRI | $F^{p \neq q}$ | 0.31 $\pm$ 0.2 | 0.16 $\pm$ 0.19 | 0.22 $\pm$ 0.18 | **0.49** $\pm$ 0.28 | 0.34 $\pm$ 0.13 | 0.06 $\pm$ 0.12 | 0.35 $\pm$ 0.08 |

Table 3: Results for simulated data with hidden common causes. The mean and the standard deviation of the F1 score are reported and the best results are in bold. Double-bars are used for grouping methods according to the class they belong to.

| | Perf. | FCIGCE | tsFCI | TCDF |
|---|---|---|---|---|
| $7t2h_{t>0}$ | $F^{p \neq q}$ | **0.57** $\pm$ 0.1 | 0.52 $\pm$ 0.1 | 0.02 $\pm$ 0.1 |
| $7t\mathring{2}h_{t>0}$ | $F^{p \neq q}$ | 0.33 $\pm$ 0.1 | **0.36** $\pm$ 0.1 | 0.07 $\pm$ 0.1 |
| | $F^{p=q}$ | 0.83 $\pm$ 0.1 | **0.99** $\pm$ 0.1 | 0.19 $\pm$ 0.2 |

and $0.04 \pm 0.12$).

To overcome the limitations of PCGCE when $\gamma$ increases, one may think of relying on a dimension reduction technique on the past slice (e.g., using auto-encoders) or to bootstrap the variables (with a minimal ratio with respect to the sample size). This is however beyond the scope of this paper and will be explored in future work.

# 6 CONCLUSION

We have addressed in this study the problem of inferring an extended summary causal graph from observational time series using a constraint-based approach. We argue here that extended summary graphs are a privileged representation for causal graphs; they are easier to be analyzed by experts and are more complete than summary causal graphs as they do not conflate past and present instants of time series. To deal with extended summary graphs, we have first proposed a greedy causation entropy measure which generalizes causation entropy to lags greater than one and to instantaneous relations. This measure, together with standard mutual information for instantaneous relations, is used to assess whether two time series are causally related or not. We have then shown how to adapt standard PC-based and FCI-based algorithms for extended summary graphs in time series, for (non) causally sufficient structures. Experiments conducted on different benchmark datasets and involving previous state-of-the-art proposals showed that the methods we have introduced provides a good trade-off between efficiency and effectiveness compared to other constraint-based methods.

Preliminary experiments suggest that the proposed method may loose accuracy when the time lag is important. Several strategies can nevertheless be proposed to overcome this problem, strategies that we intend to explore in the future.

# Acknowledgements

This research was partly supported by MIAI@Grenoble Alpes (ANR-19-P3IA-0003).

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
