# OpenReview forum: "Discovery of Extended Summary Graphs in Time Series"
_auai.org/UAI/2022/Conference — UAI 2022 Poster_

### Official Review · Reviewer_1MNt · 2022-04-12

**Q2(1) Originality/Novelty:** 3
**Q2(2) Significance/Impact:** 3
**Q2(3) Correctness/Technical Quality:** 3
**Q2(6) Clarity Of Writing:** 3
**Q6 Overall Score:** 7
**Q8 Confidence In Your Score:** 2

**Q1 Summary And Contributions:**

This paper proposes a causal discovery method from time-series data using information-theoretic measures to produce an extended extended summary causal graph, which can represent two types of relations: from the past to the present and instantaneous relations in the present slice.

**Q2 Assessment Of The Paper:**

More detailed information regarding each of these aspects is given below:

**Q2(4) Quality Of Experiments (Optional):**

3: Good: The experimental evaluation is adequate, and the results convincingly support the main claims.

**Q2(5) Reproducibility:**

4: Excellent: Key resources (e.g., proofs, code, data) are available and key details (e.g., proof sketches, experimental setup) are comprehensively described for competent researchers to confidently and easily reproduce the main results.

**Q3 Main Strengths:**

This paper proposes a causal method to infer an extended summary causal graph from observational time series using a constraint-based approach. This approach is novel and has not been studied before. The produced graphs are more robust that window causal graphs as they do not depend on the sampling rate used to collect data. In addition, they are more complete than summary causal graphs as they do not conflate past and present instants of time series.  Furthermore, the strengths of this study are that it reduces computation time compared to existing studies and that it increases the accuracy of the causal graphs.

**Q4 Main Weakness:**

Although the fMRI data used in this study are assumed to be real data, I believe they are artificial data created by Smiths et al. (2010).

**Q5 Detailed Comments To The Authors:**

It would be easier to understand if the bullet points summarize the advantages of the proposed method in this paper in the introduction section.
Please check if the fMRI data used in this paper are not simulated data.

**Q7 Justification For Your Score:**

The method proposed in this study is novel and its effectiveness has been demonstrated by experiments. It is highly likely that this study will lead to the widespread use of extended summary graphs in the future.

**Q9 Complying With Reviewing Instructions:**

1: Yes.

---

### Official Review · Reviewer_ygVt · 2022-04-12

**Q2(1) Originality/Novelty:** 3
**Q2(2) Significance/Impact:** 2
**Q2(3) Correctness/Technical Quality:** 3
**Q2(6) Clarity Of Writing:** 3
**Q6 Overall Score:** 5
**Q8 Confidence In Your Score:** 3

**Q1 Summary And Contributions:**

The paper introduces a novel method to learn the causal structure of time series. The method constructs an extended summary causal graph in which all delayed causal relationships between present and past of two time series are summarized. Relationships are examined via a newly introduced information criterion called the greedy causation entropy. The method is tested on both simulated and real-world data.

**Q2 Assessment Of The Paper:**

More detailed information regarding each of these aspects is given below:

**Q2(4) Quality Of Experiments (Optional):**

2: Fair: The experimental evaluation is weak: important baselines are missing, or the results do not adequately support the main claims.

**Q2(5) Reproducibility:**

3: Good: Key resources (e.g., proofs, code, data) are available and key details (e.g., proofs, experimental setup) are sufficiently well-described for competent researchers to confidently reproduce the main results.

**Q3 Main Strengths:**

The method introduced in the paper is, to my knowledge, a novel and interesting approach to combine information theory and causal discovery. It tackles one of the main challenges in causal discovery which is making algorithms more efficient. The computational improvements compared to other constraint-based methods are illustrated in the experiments.
The paper features a thorough evaluation of the method on both simulated and real-world data. The introduced method is compared to several different methods that are commonly used to search for causal relationships in time-series data. Compared to other constraint-based approaches, the performance is equivalent or better.
The paper is well written overall and easy to follow. Parameter choices are shown and code is provided in the supplementary material.

**Q4 Main Weakness:**

I have questions regarding the design of the experiments. One of the main selling points of the introduced greedy causation entropy is that it can be rewritten and computed with low computational complexity. In the simulations, all five considered causal structures have a single ‘past’ time step. This seems to negate the purpose of the greedy causation entropy. Likewise, the main advantage of extended summary causal graphs compared to ‘standard’ summary causal graphs is that relationships between time series can be either from ‘past’ to ‘present’ or be instantaneous, but the simulations do not include any settings in which both types of relationships are present simultaneously.
In the evaluation, linear state of the art methods struggle on the simulated, highly non-linear data, but perform well on the real-world data. It would have been good to include ‘more linear’ simulations to examine whether the linearity is the reason some methods perform better in the real world.

**Q5 Detailed Comments To The Authors:**

1) Introduction:
- I am not entirely sold on extended summary causal graphs. Isn't a window causal graph strictly superior, as it just includes more information and can be converted to an extended summary causal graph, but not the other way around?

4) Causal Discovery:
- To my knowledge, PC-stable is an adaptation of PC where edges are removed only after the size of the conditioning set is increased. Isn’t this different from what is applied here?

5) Experiments:
- Maybe I missed it: What is the sample size of the datasets?
- Which of the methods cannot handle 50 time series and why? I think this would be good to include.
- I guess there are cases (e.g., for FCIGCE) in which the orientation of an edge cannot be determined by the orientation rules. How are these cases taken into account in the evaluation (e.g., are they counted as incorrect in the calculation of the F1 score)?
- Minor point: Table 1 would look nicer if the connection between nodes and the nodes themselves would not collide (e.g., (a) 3).

6) Conclusion:
- Doesn’t the extended summary graph also depend on the sampling rate (as in if the sampling rate is very low, high-frequency interaction will not be recovered)?
- Typo: standard mutual inforamtion


**Q7 Justification For Your Score:**

I think that the paper is an interesting read and tackles an important problem in causal discovery from time-series data. It improves computational efficiency which is one of the main hurdles that is holding back causal discovery in practical settings. I have questions about the experimental design which potentially I do not understand correctly and hope that the authors can address these questions.

**Q9 Complying With Reviewing Instructions:**

1: Yes.

---

### Official Review · Reviewer_dB3o · 2022-04-12

**Q2(1) Originality/Novelty:** 2
**Q2(2) Significance/Impact:** 3
**Q2(3) Correctness/Technical Quality:** 4
**Q2(6) Clarity Of Writing:** 4
**Q6 Overall Score:** 7
**Q8 Confidence In Your Score:** 4

**Q1 Summary And Contributions:**

This paper introduces adaptions of PC and FCI to learning extended summary causal graphs from time-series. The adaptions are built on a notion of greedy causation entropy as a measure of influence from one time-series onto another.

**Q2 Assessment Of The Paper:**

More detailed information regarding each of these aspects is given below:

**Q2(4) Quality Of Experiments (Optional):**

3: Good: The experimental evaluation is adequate, and the results convincingly support the main claims.

**Q2(5) Reproducibility:**

4: Excellent: Key resources (e.g., proofs, code, data) are available and key details (e.g., proof sketches, experimental setup) are comprehensively described for competent researchers to confidently and easily reproduce the main results.

**Q3 Main Strengths:**

The paper is very well and instructively written. The proposed procedure and focus on learning the extended summary graph (instead of learning a window or full-time graph) is well motivated and this shifting of the goal of inference appears relevant to increase applicability and potential practical relevance.

**Q4 Main Weakness:**

It is unclear to me whether the simulated non-linear systems are per construction guaranteed to be stable (e.g., for linear autoregressive processes one could check the eigenvalues of the parameter matrix). If yes, can the authors comment and add a clarification to the paper? If no, can the authors defend their choice of simulations?

Score-based algorithms are wrongly said to come "without any guarantee" in Section 2. To correct the presentation of score-based algorithms and their theoretical underpinnings, the authors may wish to, among others, consult [1–6]. For the discussion of end-to-end continuous score-based structure learning algorithms in Section 2, [3] appears relevant.

**Q5 Detailed Comments To The Authors:**

# Minor comments

1. Figure 1 and Introduction – 2TBNs (two time-slice Bayesian networks) are a related representation of dynamic Bayesian networks (see, e.g., Koller & Friedman, Probabilistic Graphical Models, 2009).
2. Introduction, typo in "Consider full-time causal graph is not realistic".
3. Clarification question re "summary hides the temporal relations" – does the proposed extended summary causal graph also hide the temporal relations but to a lesser extend as it disambiguates instantaneous from non-instantaneous effects while the temporal component of the non-instantaneous effects is still lost compared to the window representation?
4. Footnote 2. I am not sure I understand the statement. The way I understand the footnote, it is aimed to somewhat justify the assumption of acyclicity even under time-instantaneous cause-effect relationships – or does it say something else? If not, I am not sure it is helpful as it conflates "acyclicity assumption under time-instantaneous causation" with another assumption of causal stationarity, one can have either without the other. In either case, this seems to be more clearly expressed later in Section 3, so perhaps this footnote could be dropped.
5. State once explicitly in Section 3, that those are not only standard assumptions for causal discovery but also adopted throughout in this work.
6. May consider streamlining Section 3, where in the beginning the cause is Xr, while later Xp.
7. Typo in paragraph before Section 4.3: "which corresponds _to_ the Markov equivalence class".
8. Typo in first sentence of Section 4.3: "there exist unobserved causes of two variables" or "there exist unobserved variables that cause two variables".
9. Section 5: introduce abbreviation for functional magnetic resonance imaging as not all readers may be familiar.
10. If I understand correctly, none of the experiments considers hidden variables that cause two observed time-series at different time lags. Is this intentional?
11. Typo in Section 6: more robust _than_ window causal graphs" and "inforamtion".

---

[1] Bühlmann et al. The Annals of Statistics, 2014

[2] Chen et al. Biometrika, 2019

[3] Reisach et al. NeurIPS, 2021

[4] Chickering. JMLR, 2002

[5] Loh & Bühlmmann. JMLR, 2014

[6] Park. JMLR, 2020

**Q7 Justification For Your Score:**

The paper presents a well executed contribution of adapted PC and FCI algorithms with a systematic discussion of their skeleton and orientation rules and the estimation of the underlying greedy causation entropy. Overall, I consider the weaknesses minor and easy to fix (discussion of score-based algorithms & typos) or justifiable (there is always more/other experiments one could run).

**Q9 Complying With Reviewing Instructions:**

1: Yes.

---

### Official Review · Reviewer_zq4b · 2022-04-12

**Q2(1) Originality/Novelty:** 2
**Q2(2) Significance/Impact:** 2
**Q2(3) Correctness/Technical Quality:** 2
**Q2(6) Clarity Of Writing:** 2
**Q6 Overall Score:** 5
**Q8 Confidence In Your Score:** 2

**Q1 Summary And Contributions:**

This paper considers causal structure learning in time series and approaches this by means of "extended summary graphs". These graphs involve exactly two vertices per component time series, one for the contemporaneous time step and one for the entire past. The paper presents adaptions of the PC and FCI algorithms to learning these graphs with independence tests based on mutual information and the proposed "Greedy Causation Entropy". Numerical experiments compare to a range of existing methods.

**Q10 Ethical Concerns (Optional):**

No ethical concerns.

**Q2 Assessment Of The Paper:**

More detailed information regarding each of these aspects is given below:

**Q2(4) Quality Of Experiments (Optional):**

2: Fair: The experimental evaluation is weak: important baselines are missing, or the results do not adequately support the main claims.

**Q2(5) Reproducibility:**

3: Good: Key resources (e.g., proofs, code, data) are available and key details (e.g., proofs, experimental setup) are sufficiently well-described for competent researchers to confidently reproduce the main results.

**Q3 Main Strengths:**

- The concept of extended summary graphs is very interesting and, I suppose, indeed useful for causal discovery. It is also novel to me.
- There is a comprehensive discussion of related work.
- Code is provided.

**Q4 Main Weakness:**

- Despite being written well in terms of language and structure, significant parts of the discussion and central technical aspects remain somewhat unclear to me. This concerns some definitions as well as the design the proposed algorithms. This makes it hard to evaluate formal claims.
- The numerical experiments may be missing the point, but this is not clear.

For more details on both points see my comments in Q5 below.

**Q5 Detailed Comments To The Authors:**

- The font appears to be different from that in all other submissions. Did you change the provided style file? Please do not do this. Comment to the AC: The font or fontsize used here is bigger as compared to the provided style. So I think the paper would have fit on 8 pages also with the provided style.

- "contrary to summary causal graphs, window causal graphs and extended summary causal graphs are assumed to be acyclic": Yes, although it is not really an assumption on the window causal graph or extended summary graph, right? The typical assumption is that the full time graph is acyclic. It is then a *consequence* that the associated window causal graphs and extended summary causal graphs are acyclic.

- "which states that a cause occurs before its effects": A reader may misunderstand this to assume that contemporaneous edges are disallowed, which is not the case here. You could instead write that 'effects do not occur before their causes'.

- Equation (1) and corresponding discussion: You probably also assume that the noise terms are serially and mutually independent of each other, not only independent of the causes. At least in the causally sufficient scenario.

- "which are actual causes of $X^q_t$" and one can conclude that $X^p$ does not cause $X^q$": I am not entirely sure what you mean by 'causes' and 'not causes' in these two statements. In the first statement, I suppose, you are referring to directed causes, i.e., $X^r_{t-\gamma_i} \rightarrow X^q_t$ in the full time graph if $X^r_{t-\gamma_i} \in C^q_t(X_r)$, while in the second statement, I suppose, you are including indirect causes, i.e., there is no directed path from any $X^p_s$ to any $X^q_t$, right? Please make this more precise. In general, using graphical concepts for explanations greatly aids understandability in my view.

- "left-hand side of the above equality is greater than or equal to the right-hand side": Both sides of an equation are, by definition, equal. I suppose you want to say that the left-hand side of (b) in Property 1 is greater or equal than the left-hand side of (a) in Property 1, right?

- Property 1: I am not entirely sure about part (c) of the statement, it depends on what you mean by "there is no instantaneous causal relation
between $X^p$ and $X^q$". This could mean two things. First, it could mean that there is no directed path between $X^p_t$ and $X^q_t$. Then I think the statement is incorrect, because you could have contemporaneous confounding (say, e.g, $X^p_t \leftarrow X^r_{t-\tau} \rightarrow X^q_t$ with some $\tau \geq 0$ for all $t$ and no other than these edges). Second, it could mean that also such such confounding paths are excluded. Then the statement is correct I think.

- Equation (3): This definition of conditional GCE is unclear to me. There are two possibilities for the quantity on the right-hand side, depending on whether $t^{\ast}$ is $t$ or the time window $t-\gamma:t-1$, but this choice is not reflected on the left-hand side. So when writing $GCE(X^p \rightarrow X^q|\textbf{X} ^R)$ it is unclear which of the possibilities is being meant. See also my next comment.

- "Because of Prop. 1, one can conclude that past instants of $X^p$ do not cause $X^q$ iff there exists $\mathbf{X}_R = { X^r_1 , \ldots , X^r_m}$ , with $m \geq 0$, such that $GCE(X^p \rightarrow X^q|\textbf{X} ^R) = 0$": To me this is unclear in several regards. First, the above mentioned ambiguity of what 'does not cause' means remains. Second, the ambiguity of $GCE(X^p \rightarrow X^q|\textbf{X} ^R)$ remains. Third, Property (1) does not consider *conditional* mutual information, so the relation of this claim to Property 1 is unclear.

- Regarding section 4.1:

1. From this discussion I do get a rough idea of how the algorithm works, but I am not entirely about the details. Namely: Which conditioning set is tested for which pair of variables at which step. Is it, essentially, the PC algorithm just run on the extended summary graph? It would be very helpful to have pseudocode. Without that I find it hard to confidently assert the algorithm's validity.

2. Speaking of the algorithm's validity (soundness and completeness): Since extended summary graphs are novel graphical concept, in my view this is nontrival and thus requires a more formal proof. (It might be, though, that without my confusion regarding the connection of eq. (3) and Property 1, see above, this might be more obvious.)

- Theorem 1 and corresponding discussion: Please clarify that here you are talking about the CPDAG of the extended summary causal graph.

- Definition 2: This definition seems to resemble the definition of the D-Sep sets in (Spirtes et al., 2020) but not necessarily the definition of the Possible-D-Sep sets. So I am not entirely sure whether this is correct.

- Section 4.3: How is the extended summary graph over the observed variables defined in the presence of hidden common causes? Is it, for example, the MAG implied by marginalizing out the unobserved variables in the extended summary graph over the full set of variables (including both observed and unobserved variables)? This specification is needed to discuss and evaluate whether FCIGCE is doing what it is supposed to do. In addition, also here pseudocode would be really helpful.

- "selection bias, a phenomenon that is not present in the datasets we consider": It seems worth mentioning that then there cannot be edges of the type undirected and undirected on one side and undetermined on the other.

- "analysis on a real world datasets": datasets --> dataset

- Several important comments regarding the experiments:

1. From Table 1 one may get the impression that the maximal lag $\gamma$ is $\gamma = 1$ for all synthetic datasets (when reading these graphs as window causal graphs). If this is so, then the experiments in my view miss the point because for $\gamma = 1$ the difference between window causal graphs and extended summary causal graphs disappears while the central point the paper is trying to make is that extended summary causal graphs can be more robustly estimated than window causal graphs.

2. However, when reading the graphs in Table 1 as extended summary graphs, then this concern disappears. In this case the synthetic data-generating models would be not uniquely defined, though, because an extended summary graph does not uniquely specify the full time graph.

3. The equation in the left column of page 6 does not resolve this confusion because there $\gamma$ is unspecified (other than saying $\gamma \geq 0$). Also a quick look in the provided code did not help (although it might be possible to infer when taking a closer look).

4. The authors need to clarify this important point in the discussion period.

- The discussion of the results in Tables 2, 3, and 4 seems reasonable and appropriately neutral.

- "are more robust that window causal graphs": that --> than

- "that the methods we have introduced provides": methods --> method

**Q7 Justification For Your Score:**

Given my concerns about the numerical experiments in combination with some remaining vagueness regarding technical aspects, I tend to vote for rejection. I find this quite unfortunate because I like the concept of extended summary causal graphs and am thus  eager to read the authors' reply.

--------

Update to "Borderline accept" after rebuttal and discussion.

**Q9 Complying With Reviewing Instructions:**

1: Yes.

---

### Official Review · Reviewer_zSWN · 2022-04-14

**Q2(1) Originality/Novelty:** 3
**Q2(2) Significance/Impact:** 2
**Q2(3) Correctness/Technical Quality:** 3
**Q2(6) Clarity Of Writing:** 2
**Q6 Overall Score:** 5
**Q8 Confidence In Your Score:** 4

**Q1 Summary And Contributions:**

The papers introduces the problem of learning an extended summary causal graph of time-series. The propose novel extensions of CD algorithms PC and FCI by way of generalising causal entropy.

**Q2 Assessment Of The Paper:**

More detailed information regarding each of these aspects is given below:

**Q2(4) Quality Of Experiments (Optional):**

3: Good: The experimental evaluation is adequate, and the results convincingly support the main claims.

**Q2(5) Reproducibility:**

3: Good: Key resources (e.g., proofs, code, data) are available and key details (e.g., proofs, experimental setup) are sufficiently well-described for competent researchers to confidently reproduce the main results.

**Q3 Main Strengths:**

- Interesting problem and suggested solution
- Good justification for path taken
- Time-series and causality is a understudied topic at large and it is great to see work which addresses this domain

**Q4 Main Weakness:**

- The writing is poor and the chosen notation is dense and at large difficult to understand
- Lots of typos and missing 'mathematification' of expressions - it looks like this was submitted in a hurry
- Because of the above the paper is hard to understand at times (because of how it is structured and the amount of information that appears to be left out in relevant places such as figure and table legends).

**Q5 Detailed Comments To The Authors:**

# INTRODUCTION

- It would make more sense to move figure 1 to page 1, given its extensive discussion at the first point on page one.

# Related work

- You should be very careful with talking about "Granger causality" as 'real causality' in the sense with which it is viewed in the more formal setting (e.g. Pearl, and presumably why you included it). Granger causality is an econometric test used to verify the usefulness of one variable to forecast another. Granger causality only provides information about forecasting ability, it does not provide insight into the true causal relationship between two variables - which is what you're trying to uncover in this work. The latter task is what one is usually trying to uncover with causal discovery (which has its own flaws but we won't go into that in this review).
- This is too ambiguous: "The maindrawbacks of these approaches are the need of a large sample size" - how big is large? Who considers what large?
-  And then to follow what is "good" performance?
- You say "it is however suited to discover extended summary graphs as it can consider potentially complex relations between timestamps in different time series through the use of windows" - is then the implication that the other metrics _do not_ work on extended summary graphs?

# GREEDY CAUSATION ENTROPY

- Why do you call it greedy?
- The estimation section, end of section 3.1, could really do with a figure to deconstruct the very dense notation that you have used. Whilst there is nothing difficult about the notation itself, there is a lot of it, so you will help the reader by deconstructing your exposition with a graphical depiction.
- Are there alternatives to kNN for estimating GCE? What's the difference in performance if so?

# CAUSAL DISCOVERY FOR EXTENDED SUMMARY GRAPHS

- Instead of writing the algorithm verbally as you have done in section 4.1, write it with an algorithm block instead, they are easier to follow and more compact if you are short on space. The same can be said for the top paragraph of page 6. Instead of verbally explaining the procedure; put it in an algorithm block.
- In 4.3 how do you represented a measured/observed common cause?

# Experiments

- What's the significance of using $\circ$ above $t$?
- I don't understand the naming convention for the causal structures in table 1 - what does your chosen convention actually mean?
- Typo in legend of table 1; 'commn' -> 'common' (technically the legend for a table should be above the table, not below it - they are only below for figures).
- Put your metrics used in the legend of table 2 (same things, legend goes above table). Also tell us what bold means in the table and furthermore, what we should be looking for; is the range between 0 and 1 (yes since you're using F1) and is higher better? Is lower better? Same thing for table 3.
- The proposed method wins in 2 of 5 test metrics. It draws equal with PCMCI in the third.
- It would be helpful in your tables if you explain how your ordering the methods in the tables and what your usage of double-bars in the tables actually means. It looks like you are grouping them according to type but it would be helpful if you spelled this out.
- VarLiNGAM also appears to win, as do multiple other methods, in table 3.

**Q7 Justification For Your Score:**

I am not convinced by the method given the poor performance against competing methods as shown in the experimental section (which was very good by the way). I will need additional convincing as to why the community needs another task and discovery methods, which do not seem to be performing particularly well. But alas, am happy to be proved wrong.

**Q9 Complying With Reviewing Instructions:**

1: Yes.

---

### Decision · Program_Chairs · 2022-05-15

**Decision:**

Accept (Poster)

**Comment:**

Meta Review: This paper considers causal structure learning in time series by means of "extended summary graphs" which represent each variable by two vertices, one for the contemporaneous time step and one for the entire past (up to some maximum lag). The paper presents adaptions of the PC and FCI algorithms to learning these graphs with independence tests based on mutual information and the proposed "Greedy Causation Entropy". Numerical experiments compare to a range of existing methods.

Pros:
* Idea is interesting and likely very useful for causal discovery
* Comprehensive discussion of related work.
* Code is provided.

Cons:
* Partially difficult to understand, sloppy mathematics and no pseudo-code, and complicated notation
* Lack of more targeted experiments showing strengths/weaknesses
* Real data is actually simulated data

The reviewer's vote (confidence) was 5(4) / 5(2) / 7(4) / 5(3) / 7(2). The idea presented certainly is useful, but in my view the paper needs to be improved quite a bit and is still borderline. Several of the shortcomings have been brought up by the reviewers and myself and the authors promised to work these out. Only if these are all included in the camera-ready version, I suggest to accept.

In general, simulation studies should be designed to not only show strengths, but also weaknesses. And for the proposed method there certainly is a trade-off where the proposed method looses low detection-power for high maximum time lags (gamma), while window-approaches only have increased computational time. Note that in real data there can be large true time lags and finding such lags can be computationally and statistically more effective with a window-approach. This is important for users to know.

Requirement for camera-ready version:
- simulations with both instantaneous and lagged relations: you can always evaluate TPR/FPR separately for both types, but the models should contain both since otherwise the method can be replaced by pure Causation Entropy (or others) or the pure PC algorithm (plus conditions on auto-lag).
- simulation study with increasing time lag from, say 1 to 50 given a fixed sample size.
- proper pseudo-code for all algorithms: at present it's, for example, unclear what is meant by " the conditional mutual information is used for edges in the present slice" in Sect. 4.1. What are the conditions of the CMI here?
- clean up notation
- suggestion: separately evaluate adjacency and orientation skill